# Cross-Talk between Diet-Associated Dysbiosis and Hand Osteoarthritis

**DOI:** 10.3390/nu12113469

**Published:** 2020-11-12

**Authors:** Marta P. Silvestre, Ana M. Rodrigues, Helena Canhão, Cláudia Marques, Diana Teixeira, Conceição Calhau, Jaime Branco

**Affiliations:** 1CINTESIS, NOVA Medical School, NMS, Universidade Nova de Lisboa, 1169-056 Lisboa, Portugal; claudia.sofia.marques@nms.unl.pt (C.M.); ccalhau@nms.unl.pt (C.C.); 2CHRC, CEDOC, NOVA Medical School, NMS, Universidade Nova de Lisboa, 1169-056 Lisboa, Portugal; anamfrodrigues@gmail.com (A.M.R.); helena.canhao@nms.unl.pt (H.C.); diana.teixeira@nms.unl.pt (D.T.); jaime.branco@nms.unl.pt (J.B.)

**Keywords:** diet, dysbiosis, lipopolysaccharide, hand osteoarthritis, obesity, trimethylamine-N-oxide, vitamin D

## Abstract

Hand osteoarthritis (OA) is a degenerative joint disease which leads to pain and disability. Recent studies focus on the role of obesity and metabolic syndrome in inducing or worsening joint damage in hand OA patients, suggesting that chronic low-grade systemic inflammation may represent a possible linking factor. The gut microbiome has a crucial metabolic role which is fundamental for immune system development, among other important functions. Intestinal microbiota dysbiosis may favour metabolic syndrome and low-grade inflammation—two important components of hand OA onset and evolution. The aim of this narrative is to review the recent literature concerning the possible contribution of dysbiosis to hand OA onset and progression, and to discuss the importance of gut dysbiosis on general health and disease.

## 1. Introduction

Osteoarthritis (OA) is a heterogeneous disease with a complex etiology, often associated with aging, trauma, genetic predisposition, and obesity-related metabolic dysfunction [1,2]. Despite the complex definition, little is known about the underlying mechanisms involved in the disease pathophysiology, and consequently OA is a growing public health burden: the disease prevalence is increasing dramatically (+35% between 1990 and 2015), such that OA has been classified as the non-communicable disease causing the highest rates of age-standardized disability-adjusted life-years (DALY) [3]. It may seem reasonable to interpret these observations as a consequence of world population aging, however, this reason alone does not offer an entire explanation for these increased rates [4]. Hand OA is the most prevalent form of the disease [5], but still, there is an enormous gap between the guidelines for its management and the current standards of treatment [6,7,8]. Such gap may be a result of the limited amount of evidence-based knowledge on the specific pathophysiology of hand OA, the molecular mechanisms involved, and how it differentiates from OA at other joint sites.

Hand OA is a heterogeneous condition, often involving multiple joints, and can have distinct (but sometimes overlapping) patterns of joint involvement: for example, OA of the interphalangeal joints (IPJs), and/or the first carpometacarpal joint (CMCJ). Hand OA predominates in women. The disease’s clinical manifestations frequently begin in the peri-menopause period, causing articular pain, stiffness, swelling, and limitation function, sometimes as severe as in rheumatoid arthritis [9]. Several factors have been associated with hand OA incidence and progression, namely obesity and metabolic syndrome [10]. However the pathophysiologic pathways are still poorly understood due to the barriers researchers find in the study of hand OA pathogenesis. Firstly, researchers have limited access to diseased tissue, and when available, the quantities of tissue obtained for molecular analysis are small. Secondly, there are no animal models of hand OA. An important question to be addressed is whether hand OA shares similar pathogenic pathways with OA at other joint sites, namely the inflammation caused by metabolic syndrome and favored by dysbiosis, as stated by the European Society for Clinical and Economic Aspects of Osteoporosis, Osteoarthritis and Musculoskeletal Diseases (ESCEO) [11].

## 2. Pathogenesis of Hand OA—Current Evidence

The activation of the innate immune system may be involvedin the initiation and perpetuation of OA [12], including hand OA. This activation appears to involve the macrophage-associated inflammatory response [13], activation of toll-like receptor (TLR) pathways [14], complement factors [15], and activation of the coagulation pathway (an intrinsic effector of innate immunity) [16]. Nevertheless, studying the mechanisms involved in the development of hand OA is difficult for the reasons mentioned above: limited access to diseased tissue and lack of animal models of hand OA. Consequently, it is still unclear whether the hand shares similar pathogenic pathways with OA at other joint sites.

There is evidence to suggest the involvement of mechanical loading in the development of hand OA, mainly due to an increased prevalence of the disease in the dominant hand when compared to the non-dominant hand [17]. Mechanical load influences growth factor bioavailability, inflammation, and matrix degradation [18]. Inflammation per se also appears to be part of the disease pathophysiology [19], as it is for other OA joint sites. A wide range of inflammatory markers has been characterised in hand OA, providing insight into disease development [20,21]. These include, among others, C-reactive protein, adipose tissue cytokines (e.g., adiponectin, resistin, and visfatin), and markers of cartilage or bone homeostasis (e.g., type II collagen). When comparing radiological assessments of erosive and non-erosive hand OA with healthy controls, levels of resistin and visfatin were significantly higher in the hand OA patients (both erosive and non-erosive) than in the healthy controls. In the same study, visfatin was significantly higher in erosive hand OA when compared to non-erosive hand OA or controls [21]. Moreover, in a recent small randomized controlled trial (RCT) undertaken in 18 patients, serum interleukin-1 (IL-1) levels were associated with loss of hand function and radiological erosions [22]. Although these studies suggest that different biomarker profiles could identify different severities of hand OA, current biomarker characterization in hand OA is limited and remains poorly understood. Unlike other joint sites, the early inflammatory phase of hand OA seems to pre-date bone remodeling [23]. Other specific characteristics of hand OA are the strong relationship between heritability and disease prevalence (hand OA has the highest estimated heritability of all types of OA (approximately 60%) [24]), and the fact that symptoms often occur around the time of menopause [25], suggesting a role for sex hormones in disease pathogenesis.

## 3. Relationship between Hand OA and Obesity

Obesity has been well described as one of the main risk factors for OA in weight-bearing joints, such as knee and hip joints [26,27,28], but evidence suggests that this relationship is not limited to the obesity-induced increased biomechanical load. Overweight and obese individuals also have an increased risk for OA in hands [28], suggesting that systemic factors may also be involved in this relationship. Indeed, excessive adiposity, particularly visceral adiposity, and its consequent metabolic overload leads to a vicious cycle of chronic low-grade inflammation [29]—a well-known feature of the OA pathophysiology, often preceded by an aberrant joint tissue metabolism and several anatomical and/or physiological derangements [1]. Low-grade inflammation in patients with OA is widespread, both locally and, when joint disease is generalised, systemically [30]. Ultimately, this scenario culminates in the loss of normal joint function [1]. In line with this, some authors suggest that the growing prevalence of OA, including the most prevalent hand OA, is likely to be a result of the parallel increases in the rates of obesity and chronic metabolic inflammation (metaflammation), favored by a sedentary and nutritionally poor lifestyle [31].

Notably, sarcopenia, which is the loss of muscle mass and strength or physical function, synergistically worsens the adverse effects of obesity in older adults, including OA [32]. Interestingly, while the positive effect of exercise in hip and knee OA has been documented, the effect of exercise on hand OA remains uncertain. A recent meta-analysis of clinical trials which compared therapeutic exercise versus no exercise on hand OA concluded that there is only low-quality evidence showing a small beneficial effect of exercise on hand pain, function, and finger joint stiffness [33]. In this meta-analysis, the number of selected studies was small (five clinical trials), the number of participants was limited, and the confidence intervals were wide for the outcomes of pain, function, and joint stiffness. Indeed, there is evidence supporting a negative role of hand OA on grip strength [34], but the opposite relationship—the role of grip strength on the hand OA development—has not been studied.

### Obesity-Related Gut Dysbiosis

The gut microbiome varies enormously from one individual to another, and is also highly dynamic, being under the influence of several of factors such as age, geography [35], hormones [36], and most importantly, diet [37].

Research on a potential cross-talk between obesity and the gut microbiome started over a decade ago [38] and a well-established relationship between obesity-associated metabolic abnormalities and gut dysbiosis has evolved [39,40,41]. The most well-characterised pathways in this relationship include bacterial-derived metabolites [42,43,44] and the bile acid metabolism [45,46]. The increased ratio of Firmicutes to Bacteroides observed in obese individuals and animal models of obesity (ob/ob mice) cause a higher production of biologically active metabolites such as short-chain fatty acids (SCFAs) from soluble dietary fibers (i.e., fructans) and resistant starch, which leads to an increased energy harvest from otherwise indigestible carbohydrates [24,25]. Consequently, greater adipogenesis may occur in the liver. Research undertaken in animal models of obesity has brought similar evidence over the years: a high-fat diet changes the composition of the murine gut microbiota, reducing the levels of some Gram-positive and Gram-negative species [47]. Germ-free mice with no intestinal microbiota that are fed a high-fat diet do not develop inflammation [48]. Moreover, modulation of the gut microbiome in mice who have had microbiome-induced metabolic syndrome (MetS) due to excessive antibiotic exposure attenuated cartilage damage [49].

To date, leading theories about the mechanisms relating gut microbiome to metabolic disturbances include changes in molecular signalling molecules released by bacteria in contact with local tissue or distant organs such as the brain (gut–brain axis). Bacteria and their metabolites might target the brain directly via vagal stimulation, or indirectly through immune-neuroendocrine mechanisms [50]. The vagal nerve transmits information from enteral content to the nucleus tractus solitaries, where the information is then distributed to the hypothalamus, regulating appetite, food intake, and energy expenditure. Activation of the vagus nerve is partly dependent upon the secretion of chemical signals such as gut peptide YY (PYY), glucagon-like peptide 1 (GLP-1), and CCK by enteroendocrine cells [50]. Additionally, the gut microbiome can stimulate the reprogramming of gene expression in the colon [51]. Fasting-induced adiposity factor (Fiaf), a circulating lipoprotein lipase (LPL) inhibitor whose expression is normally suppressed in the gut epithelium by microbiota, plays a central role in triglyceride metabolism [52] by inhibiting LPL production in adipose tissue and modulating fatty acid oxidation. Specific components of the microbiota might suppress Fiaf in the intestinal epithelia, and potentially stimulate host weight gain by impairing triglyceride metabolism and promoting fat storage [53].

Considering the knowledge on the contribution of dysbiosis to obesity and related metabolic abnormalities, it is sensible to believe that there is a link between intestinal microbiota and other chronic inflammatory states, such as OA.

## 4. Gut-Dysbiosis-Derived Inflammatory Mechanisms

The numerous risk factors that are shared by gut dysbiosis and OA (e.g., aging, gender, obesity), and the potential involvement of gut dysbiosis on articular auto-immune, pro-inflammatory disease (e.g., rheumatoid arthritis) [54], suggest a possible role of the gut microbiota in the pathogenesis of OA, with hand OA being less studied but potentially highly influenced by these relationships.

Some authors have attempted to explain the involvement of gut microbiota or, more specifically, its metabolic products in the development of OA [11].

### 4.1. Metabolic Endotoxemia

Lipopolysaccharide (LPS, also known as endotoxin) is a pro-inflammatory component of the outer-membrane of Gram-negative bacteria, released in constitutively produced vesicles. These outer-membrane vesicles contain proteins that are insensitive to proteases, suggesting that they can transport molecules systemically via biological fluids and deliver them in a concentrated manner [55]. Low-grade inflammation, a common feature for obesity and OA, can result from the intestinal absorption of LPS [56]. LPS binds to its pattern recognition receptor, TLR4 [57], expressed on the cell surface of monocytes and other immune cells, as well as onvarious other cell types such as skeletal muscle, adipose tissue, and liver [58,59,60]. Binding to the TLR4 causes activation of the innate immune system through CD14 signaling [61], which has a role in the pathogenesis of OA, as described earlier in this review. In the unaffected individuals, expression of TLRs is increased in OA, and this activation leads to increased levels of nuclear factor κB (NFκB) [62]. Elevated NFκB levels are known to induce activated joint cells to produce catabolic cytokines and chemokines, such as tumour necrosis factor-alpha (TNF-α), interleukin 1-beta (IL-1β), IL-6, receptor activator of NFκB, which in turn can increase the production of matrix metalloproteinases (MMPs), decrease collagen and proteoglycan synthesis, and further augment NFκB activation. Ultimately this culminates in secondary inflammation in tissues [63]. Moreover, macrophages and neutrophils activated by LPS synthetize free radicals, leading to tissue damage [64]. Compared with a normal diet, a high-fat diet significantly reduces the numbers of the Gram-positive *Bifidobacterium* spp., which have been associated with reductions in intestinal LPS levels [65]. On the other hand, introduction of prebiotic fibre to a high-fat diet specifically increases bifidobacterial numbers and reduces plasma LPS levels, compared with an unsupplemented high-fat diet [66]. Notably, many of the mechanisms now identified as having roles in the pathogenesis of OA overlap with the immune-activating functions of lipopolysaccharide.

Alkaline phosphatase (AP), a gate-keeper of innate immune system responses, appears to have an important role in modulating the inflammatory effect of LPS thought to be involved in OA pathogenesis, at least in animal models of the disease [67]. Asan ectophosphatase, AP acts extracellularly by dephosphorylating, and consequently, detoxifying inflammation-triggering moieties (ITMs) from external and internal sources. LPS is a known ITM from an internal source (microbiota), and it is a target of AP dephosphorylation and consequent inactivation [68,69]. Exogenous ectophosphatase interventions by AP in animal models resulted in near-complete inhibition of systemic tumor necrosis factor-alpha (TNF-α), IL-6, and IL-8 response after a systemic inflammatory insult with LPS [69,70].

Huang and Kraus proposed a hypothesis in which LPS in the bloodstream would have a role in a two-hit model of OA pathogenesis and potentiation [71]. One factor would be the triggering of the proinflammatory innate immune response by LPS; the second would involve complementary mechanisms such as joint injury and damage, which synergistically activate innate immunity. The authors recently published data where the hypothesis was tested. They transplanted faecal microbiota from healthy and from metabolically compromised human donors into germ-free mice, 2 weeks before articular knee injury. Compared with the other groups, transplantation with the microbiome from metabolically compromised donors was associated with higher mean systemic concentrations of inflammatory biomarkers, higher gut permeability, and worse OA severity [72].

### 4.2. Microbiota-Derived Metabolites

Trimethylamine-N-oxide (TMAO) is a microbiota-generated metabolite derived from choline and carnitine, which are essential nutrients contained in many foods, including red meat, eggs, and dairy. Choline has a wide range of biological activities; it maintains the structural integrity of cell membranes, supports cholinergic neurotransmission, and donates methyl groups in many biosynthetic reactions [73]. Carnitine transports long-chain fatty acids into the mitochondria to produce energy. Bacteria in the human gut possess the trimethylamine lyase system (CutC/D) and the carnitine Rieske-type oxygenase/reductase system (CntA/B and YeaW/X) for metabolizing choline and carnitine from the diet into trimethylamine (TMA) [74]. Once in the liver, TMA is oxidized by flavin-containing monooxygenases (FMOs) to TMAO [75]. TMAO is an established risk factor for cardiovascular disease [76,77,78,79]. Patients who had major adverse cardiovascular events had significantly higher baseline levels of TMAO than those who did not, and TMAO levels were associated with 3.4-fold increase in mortality risk [80]. Mechanistically, TMAO appears to exert its effects on atherosclerosis/cardiovascular disease by regulating lipid metabolism and inflammation. In terms of lipid metabolism, the gut-derived metabolite was shown to inhibit reverse cholesterol transport in mice [81]. Regarding inflammation, several mechanisms have been suggested, of which some may be relevant to the current review. Primarily, TMAO can activate the NFκB signalling cascade in primary human aortic endothelial cells and vascular smooth muscle cells [82]. TMAO may also increase TNF-α and IL-1β levels and decrease anti-inflammatory factor IL-10 levels [83]. Finally, TMAO can significantly trigger oxidative stress [84]. As for LPS, TMAO mediates inflammatory processes that are part of the pathophysiology of OA, including hand OA. Despite the important role of the diet (higher versus lower meat intake) in determining systemic concentrations of TMAO [85]; these are also influenced by genetics and other host characteristics [86], with age being an important risk factor for elevated TMAO levels [85].

In 2015, the Osteoarthritis Research Society International (OARSI) highlighted the need for clinical trial evaluation of soluble biomarkers for the prediction and monitoring of hand OA [87]. Clinical research is needed to investigate the role of TMAO and LPS as potential exploratory biomarkers for better understanding and predicting hand OA.

## 5. Gut Permeability

The intestinal mucosa constitutes a selectively permeable barrier between the blood and the intestinal lumen. The intercellular tight junctions between enterocytes regulate intestinal permeability in response to physiological or pathological stimuli [86]. Experimental data in animals indicate that the intestinal barrier function is regulated in vivo through activation of the intestinal cannabinoid type 1 receptor (CB1R) [88]. Most interestingly, the gut microbiota, specifically through LPS—and possibly also nutrients—contribute to the regulation of the intestinal barrier via setting the tone of the intestinal endocannabinoid system [89]. In obese mice, CB1R is upregulated, and treatment with a CB1R antagonist results in reduced translocation of bacterial antigens such as LPS into the systemic circulation [90]. High dietary intake of fatty acids can also increase endocannabinoid levels in different tissues, including the intestine [91], suggesting a mechanism whereby a high-fat diet can increase LPS, thus triggering the development of chronic inflammation in a positive feedback loop.

## 6. Diet: Role in Dysbiosis and OA

Diet has a major impact on the gut microbiome. Besides their direct effect on gut microbial diversity and composition [92,93], dietary constituents (nutrients, phytochemicals, and others) may disrupt the protective functions of the intestinal barrier in ways that could affect the host–microbiome interface, leading to dysbiosis and consequently to inflammatory processes with clinical implications on the host. High-fat diets [94], as discussed earlier in this review, Western-style diets (high in sugar and fat) [95], or diets poor in fiber [96] have been suggested to disrupt barrier function in mice, affecting the concentration of bacterial metabolites entering the circulation. More specifically, mice harboring a balanced tissue omega-6:omega-3 polyunsaturated fatty acid (PUFA) ratio showed increased production and secretion of intestinal AP, which suppresses LPS-producing gut bacteria, such as Proteobacteria [97]. While evidence for the beneficial effects of dietary-lipid modification (increased omega-3:omega-6 PUFA) on OA is currently limited, some authors recommend supplementation with fish oil, as this will at least benefit metabolic health [98]. Additionally, there is now evidence that relatively low doses of selected emulsifiers (detergent-like molecules that are a ubiquitous component of processed foods) can erode the host’s protective epithelial mucous layer and lead to dysbiosis-mediated low-grade inflammation in experimental models [99]. Research on the emergent role of vitamin D in dysbiosis-related inflammation is a growing field and is discussed in more detail in the next section of this review.

## 7. Obesity-Related Vitamin D Deficiency: Role in Dysbiosis and OA

Vitamin D deficiency or insufficiency affects 30–60% of the population worldwide and is increasingly found in association with many pathological conditions, including obesity, MetS, and autoimmune diseases [100]. In addition to its classical roles in promoting calcium and phosphorus absorption, vitamin D in its active format—25-hydroxyvitamin D3 (25OHD3, calcitriol)—functions as a sterol hormone to regulate diverse biological functions, ranging from the host immune response to cell differentiation [101]. Although, a nutritionally poor, energy-dense diet, including a high-fat diet (HFD), is thought to be a major cause of obesity-related MetS [102,103,104], epidemiologic evidence shows that vitamin D deficiency is an independent risk factor for MetS in the elderly [105]—the age group mainly affected by hand OA. In a study conducted by Su and colleagues, plasma LPS levels were significantly increased in the mice fed a HFD or those with vitamin D deficiency (VDD), but synergistically elevated in HFD + VDD mice. Conversely, dietary 25OHD3 supplementation significantly attenuated the plasma endotoxin levels being exerted by HFD feeding [106]. Additionally, vitamin D is required for gut mucosal immune defense against pathogens and the sustenance of beneficial commensals; vitamin-D-deficient mice exhibited diminished expression of Paneth cell defensins, tight junction genes, and mucin 2 (MUC2) [106]. On the other hand, *Akkermansia muciniphila*, a symbiotic bacterium in the phylum Verrucomicrobia, was decreased significantly in the ileum of mice with VDD, or those fed on a HFD [106].

## 8. Mediterranean Diet–Modulation of Gut Microbiome

The Mediterranean Diet (MD) pattern has been linked to increased longevity and reduced morbidity [107]. The definition of the MD is still ambiguous. Nevertheless, there is consensus that it is associated with a greater consumption of vegetables, fruit, legumes, nuts, seeds, and whole-grain cereals (leading to increased fiber intake); greater intakes of fish and seafood; moderate consumption of dairy products, poultry, and eggs; frequent, albeit moderate, intake of red wine; and virgin olive oil as the main source of dietary fat [108,109,110,111]. There is evidence, albeit scarce, showing that the MD may protect against OA due to its anti-inflammatory effect, anti-obesity effect, and the antioxidant capacity of the dietary pattern [112]. However, the limited number of studies investigating these relationships have failed to address the microbiota-derived metabolites, leaving an important gap in the explanatory mechanisms of how the MD may improve OA. Considering that the “ingredients” of the MD closely interact with the gut microbial community, one may suggest that the mechanisms by which the MD protects against OA may also be mediated by an interplay between the diet and the gut microbiome. Indeed, other authors have suggested that the MD confers a beneficial modulation of the gut microbiota by (a) increasing bacterial diversity and (b) reducing metabolic endotoxemia [113].

## 9. Conclusions and Future Considerations

Obesity and related metabolic disorders, often resulting from long-term overnutrition and poor lifestyle, have been suggested to have a potential role in the development of OA, including hand OA. However, the mechanisms by which metabolic abnormalities may cause hand OA are a matter of debate. Chronic low-grade inflammation may be the link between the two conditions, often characterized as the immunometabolism. As discussed in this review, obesity-derived gut dysbiosis leads to increased concentrations of circulating pro-inflammatory molecules such as LPS and TMAO, which could explain the mechanistic relationship between obesity and hand OA (Figure 1). Another risk factor for the aberrant immunometabolism is vitamin D deficiency. A beneficial dietary manipulation of gut integrity, easily achieved by following a Mediterranean dietary pattern (rich in phytochemicals, unsaturated fats, and vitamin-D-rich foods), may help to prevent a pro-inflammatory state, hence preventing the development of associated abnormalities such as hand OA.

With the current lack of effective treatment, and the common misconception that OA of the hands does not strongly impact quality of life (when compared to OA of other limbs), there is an urgent need for novel research on the specific mechanisms behind the pathophysiology of hand OA, allowing for evidence-based approaches in prevention, management, and treatment of the disease.

## Figures and Tables

**Figure 1 nutrients-12-03469-f001:**
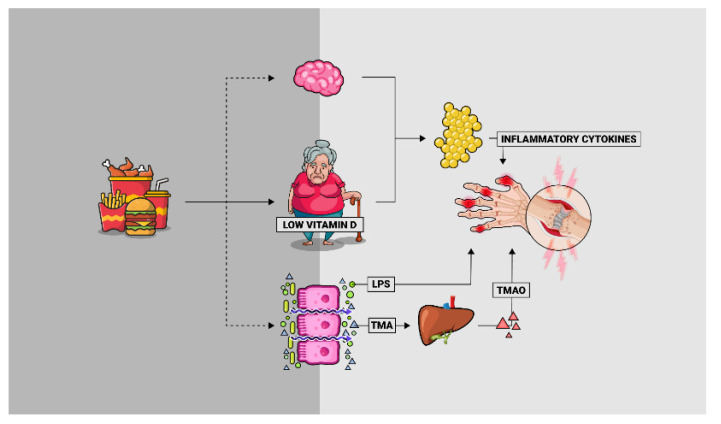
Cross-talk between diet-associated dysbiosis and hand osteoarthritis pathophysiology. A poor lifestyle, coupled with an inadequate energy balance over the years, causes increased adiposity and obesity-derived gut dysbiosis. The permeable gut, colonized by an unfavorable microbiota, leads to increased concentrations of circulating pro-inflammatory molecules such as LPS and metabolites such as TMAO that could have an impact on the aberrant inflammatory process which characterizes hand osteoarthritis. Additionally, the growing adipose tissue secretes pro-inflammatory cytokines that negatively modulate this process. LPS: lipopolysaccharide; TMA: trimethylamine; TMAO: trimethylamine-N-oxide.

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
