# Peer review of "Cross-Talk between Diet-Associated Dysbiosis and Hand Osteoarthritis"

_nutrients, 2020, doi:10.3390/nu12113469_

Round 1
Reviewer 1 Report
Title: Crosstalk Between Diet-Associated Dysbiosis and Hand Osteoarthritis
Authors: Marta P Silvestre, Ana M Rodrigues, Helena Canhao, Claudia Marques, Diana Teixeira, Conceicao Calhau and Jaime C Branco
Overview: In this manuscript the authors provide a narrative review of the recently published literature on the possible contribution of gut dysbiosis to the development of hand osteoarthritis (OA).
Specific comments: This is an interesting and timely review and I enjoyed reading it very much. However there are some suggestions for minor revisions and improvement which are outlined below.
- The authors use the abbreviation hOA for hand OA but I am not sure that this is a widely used abbreviation. I would strongly discourage the authors from using abbreviations that are not used widely. Furthermore, in some parts of the text the authors use hOA and in other parts they use hand OA. Consistency is important and I would strongly recommend that hand OA is used instead of hOA.
- On page 2, line 48 the guidelines of ESCEO are mentioned. If the authors are going to cite papers from major societies, then, in fairness, the relevant papers from EULAR, ACR and OARSI should also be cited. It is important to provide a balanced view of the literature, not just the view of ESCEO.
- Page 4, lines 143-144: please provide a citation for the sentence: "Some authors have attempted to explain the involvement of gut microbiota or, more specifically, its metabolic products in the development of OA". This sentence is missing the relevant citation.
- In line 175 on page 4, please include the Roman symbol for α in TNF-α.
- Page 5, section 6, line 222: In the sentence where the authors are discussing the role of polyunsaturated fatty acids, it would be great if they could cite our work in this area: Thomas S, Browne H, Mobasheri A, Rayman MP. What is the evidence for a role for diet and nutrition in osteoarthritis? Rheumatology (Oxford). 2018 May 1;57(suppl_4):iv61-iv74. doi: 10.1093/rheumatology/key011.
PMID: 29684218 - In the conclusion section, the authors may wish to consider the possibility that dietary modulation, the reduction in dietary consumption of animal fats, increasing the dietary consumption of olive oil and the increased intake of dietary fibre could be used to modulate the abundance of beneficial bacteria in the microbiome. The authors do not cite any of the papers that relate changes in the micro biome following dietary intervention in preclinical animals. They may consider that this area is beyond the remit of the paper but this is an important point that needs to be highlighted.
- Finally, the authors may wish to consider including a figure in this paper. Mini-reviews that include a figure and a graphical abstract are far more likely to be downloaded, read and cited.
Author Response
Dear reviewer,
Thank you for your review and efficiency in sending it within only a few days following our submission. We believe that your input will make a significant positive contribution to our publication. We have addressed all your concerns in detail. Please see our responses to each point, below.
Overview: In this manuscript the authors provide a narrative review of the recently published literature on the possible contribution of gut dysbiosis to the development of hand osteoarthritis (OA).
Specific comments: This is an interesting and timely review and I enjoyed reading it very much. However there are some suggestions for minor revisions and improvement which are outlined below.
RESPONSE: We thank you for your kind comments.
- The authors use the abbreviation hOA for hand OA but I am not sure that this is a widely used abbreviation. I would strongly discourage the authors from using abbreviations that are not used widely. Furthermore, in some parts of the text the authors use hOA and in other parts they use hand OA. Consistency is important and I would strongly recommend that hand OA is used instead of hOA. RESPONSE - We agree that hOA is not widely used and, in fact, may lead to misinterpretations (e.g. "h" could be confounded with "hip"). We have removed all acronyms, replacing it with "hand OA".
- On page 2, line 48 the guidelines of ESCEO are mentioned. If the authors are going to cite papers from major societies, then, in fairness, the relevant papers from EULAR, ACR and OARSI should also be cited. It is important to provide a balanced view of the literature, not just the view of ESCEO. RESPONSE: We absolutely agree with the reviewer. We have added citations from EULAR, ACR and OARSI to our manuscript (highlighted in yellow in text), where appropriated: Refs 7, 8 for EULAR and ARC, respectively (discussing treatment guidelines) and ref. 89 (discussing the need for clinical trials evaluating soluble biomarkers of hand OA). We have included a sentence based on reference 89 to our section 4 of the manuscript (also highlighted in yellow)
- Page 4, lines 143-144: please provide a citation for the sentence: "Some authors have attempted to explain the involvement of gut microbiota or, more specifically, its metabolic products in the development of OA". This sentence is missing the relevant citation. RESPONSE - Citation provided: Ref 56 - highlighted in yellow in text.
- In line 175 on page 4, please include the Roman symbol for α in TNF-α. RESPONSE: Corrected, we believe that this was a format issue when submitting a word version of the manuscript. We hope the error does not persist.
- Page 5, section 6, line 222: In the sentence where the authors are discussing the role of polyunsaturated fatty acids, it would be great if they could cite our work in this area: Thomas S, Browne H, Mobasheri A, Rayman MP. What is the evidence for a role for diet and nutrition in osteoarthritis? Rheumatology (Oxford). 2018 May 1;57(suppl_4):iv61-iv74. doi: 10.1093/rheumatology/key011.
PMID: 29684218 RESPONSE: We thank you for sharing your valuable work. We have added a sentence from your review (highlighted in yellow) and attributed it the reference 101. - In the conclusion section, the authors may wish to consider the possibility that dietary modulation, the reduction in dietary consumption of animal fats, increasing the dietary consumption of olive oil and the increased intake of dietary fibre could be used to modulate the abundance of beneficial bacteria in the microbiome. The authors do not cite any of the papers that relate changes in the micro biome following dietary intervention in preclinical animals. They may consider that this area is beyond the remit of the paper but this is an important point that needs to be highlighted. RESPONSE - We left our concluding paragraph free of citations as we wanted to express our critical point of view in this section. However, we thank you for your pertinent observation and agree to it. Consequently, we have added a paragraph/section, prior to conclusions, in order to discuss the potential importance of dietary manipulation on the microbiome, as suggested. The section is highlighted in yellow.
- Finally, the authors may wish to consider including a figure in this paper. Mini-reviews that include a figure and a graphical abstract are far more likely to be downloaded, read and cited. RESPONSE - We absolutely agree with the figure inclusion and, as a matter of fact, we submitted a figure representing the take home messages from our narrative review (PNG format, attached to the manuscript). A kind of graphical abstract. It was submitted in a format accepted by the journal, so we do not understand why hasn't this come across to you. We have included the figure (as print screen) in our revised word document (attached). The only reason why we haven't submitted as PDF was to allow editing for inclusion in the journal.

Reviewer 2 Report
The title reflects the subject of the study. This manuscript presents a clear and clinically useful message. It is well written in terms of clarity, style, and use of English language. All subjects are sufficiently detailed. Each section explains adequately the purpose of this narrative in the context of published information. The conclusions accurately and clearly explain the main findings. The length of the manuscript is ideal. All references are appropriate and current.
Author Response
Dear Reviewer,
We thank you very much for your review and efficiency in sending it within only a few days after the submission. We are really pleased to know that you enjoyed reading our work.
We have requested for a proof-reading and have now submitted a final version (attached).
many thanks
